# CRAMER-WOLD AUTOENCODER

## ABSTRACT

Assessing distance between the true and the sample distribution is a key component of many state of the art generative models, such as Wasserstein Autoencoder (WAE). Inspired by prior work on Sliced-Wasserstein Autoencoders (SWAE) and WAE-MMD (WAE using maximum mean discrepancy based distance function) we construct a new generative model – Cramer-Wold AutoEncoder (CWAE). CWAE cost function is based upon a characteristic kernel we introduce, called Cramer-Wold kernel, and has a a simple closed-form in the case of normal prior. As a consequence, while simplifying the optimization procedure (no need of sampling necessary to evaluate the distance function in the training loop), CWAE performance matches quantitatively and qualitatively that of WAE-MMD and often improves upon SWAE.

## 1 INTRODUCTION

One of the crucial aspects in construction of generative models is devising effective method for computing and minimizing distance between the true and the model distribution. Originally in Variational Autencoder (VAE) (Kingma & Welling, 2014) this computation was carried out using variational methods. An important improvement was brought by the introduction of Wasserstein metric (Tolstikhin et al., 2017) and the construction of WAE-GAN and WAE-MMD models, which relax the need for variational methods. WAE-GAN requires a separate optimization problem to be solved to approximate the used divergence measure, while in WAE-MMD the discriminator has the closed-form obtained from a characteristic kernel, i.e. one that is injective on distributions Muandet et al. (2017). A recent contribution to this trend of simplifying the construction of generative models is Sliced-Wasserstein Autoencoder (SWAE, Kolouri et al. (2018)), where a significantly simpler AutoEncoder based model based on Wasserstein distance is proposed.

The main innovation of SWAE was the introduction of the sliced-Wasserstein distance – a fast to estimate metric for comparing two distributions, based on the mean Wasserstein distance of one-dimensional projections. However, even in SWAE there is no close analytic formula that would enable computing the distance of the sample from the standard normal distribution. Consequently in SWAE two types of sampling are needed: (i) sampling from the prior distribution and (ii) sampling over one-dimensional projections.

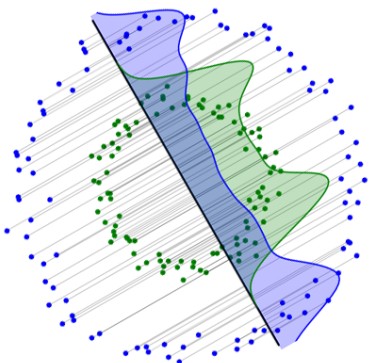

Figure 1: Cramer-Wold distance of two sets obtained as the mean squared $L_2$-distance of their smoothed projections on all one-dimensional lines.

Our main contribution is introduction of the Cramer-Wold distance between distributions, *which has a closed-form for the distance of a sample from standard multivariate normal distribution*. Its important feature is that it is given by a characteristic kernel which has a closed-form given by equation 7 for the product of radial Gaussians[1]. We use it to construct an AutoEncoder based generative model, called Cramer-Wold AutoEncoder (CWAE), in which the cost function, for a normal prior distribution, has a closed analytic formula. Thus

---

[1]For more information we refer the reader to Appendix, Sections A and B.

CWAE can be seen as a borderline model between SWAE and WAE-MMD, as we use a characteristic kernel to discriminate distributions, but its formula comes from the application of the sliced approach. An introduction of sliced approach is given in the following section, the comparison of CWAE and WAE-MMD is given in Appendix, Section B. We benchmark CWAE against generative models using a non-parametrized distance functions: WAE-MMD (WAE variant from Tolstikhin et al. (2017) based on the classical MMD distance) and SWAE. Our results show that, while simplifying the optimization procedure by forgoing the need to sample data projections or from the prior distribution we obtain an AutoEncoder based generative model which obtains quantitatively and qualitatively similar results to WAE-MMD and SWAE models, see Section 5.

Let us now briefly describe the outline of the paper. In the following two sections we introduce and theoretically investigate the Cramer-Wold distance. However, the reader interested mainly in the construction of our generative AutoEncoder model can proceed directly to Section 4. Section 5 contains experiments. Conclusions are given in Section 6.

## 2 CRAMER-WOLD DISTANCE: CONSTRUCTION

Motivated by the prevalent use of normal distribution as prior in modern generative models, we investigate whether it is possible to simplify optimization of such models. As the first step towards this, in this section we introduce Cramer-Wold distance, which has a simple analytical formula for computing normality of a high-dimensional sample. On a high level our approach uses the traditional $L_2$ distance of kernel-based density estimation, computed across multiple single-dimensional projections of the true data and the output distribution of the model. We base our construction on the following two popular tricks of the trade:

**Sliced-based decomposition of a distribution:** Following the footsteps of Kolouri et al. (2018); Deshpande et al. (2018), the first idea is to leverage the Cramer-Wold Theorem (Cramér & Wold, 1936) and Radon Transform (Deans, 1983) to reduce computing distance between two distributions to one dimensional calculations. For $v$ in the unit sphere $S_D \subset \mathbb{R}^D$, the projection of the set $X \subset \mathbb{R}^D$ onto the space spanned by $v$ is given by $v^T X$ and the projection of $N(m, \alpha I)$ is $N(v^T m, \alpha)$. Cramer-Wold theorem states that two multivariate distributions can be uniquely identified by their all one-dimensional projections. For example, to obtain the key component of SWAE model, i.e. the sliced-Wasserstein distance between two samples $X, Y \in \mathbb{R}^D$, we compute the mean Wasserstein distance between all one-dimensional projections:

$$d_W(X, Y) = \int_{S_D} d_W(v^T X, v^T Y) \, d\sigma_D(v), \tag{1}$$

where $S_D$ denotes the unit sphere in $\mathbb{R}^D$ and $\sigma_D$ is the normalized surface measure on $S_D$. This approach is effective since the one-dimensional Wasserstein distance between samples has the closed form, and therefore to estimate (1) one has to sample only over the projections.

**Smoothing distributions:** Using the sliced-based decomposition requires us to define distance between two sets of samples, in a single dimensional space. To this end we will use a trick-of-trade applied commonly in statistics in order to compare samples or distributions which is to first *smoothen* (sample) distribution with a Gaussian kernel. For the sample $R = (r_i)_{i=1..n} \subset \mathbb{R}$ by its smoothing with Gaussian kernel $N(0, \gamma)$ we understand

$$\mathrm{sm}_\gamma(R) = \tfrac{1}{n} \sum_i N(r_i, \gamma),$$

where by $N(m, S)$ we denote the one-dimensional normal density with mean $m$ and variance $S$. This produces a distribution with regular density, and is commonly used in kernel density estimation. If $R$ comes from the normal distribution with standard deviation close to one, the asymptotically optimal choice of $\gamma$ is given by the Silverman's rule of thumb $\gamma = (\frac{4}{3n})^{2/5}$, see Silverman (1986). For continuous density $f$, its smoothing $\mathrm{sm}_\gamma(f)$ is given by the convolution with $N(0, \gamma)$, and in the special case of Gaussians we have $\mathrm{sm}_\gamma(N(m, S)) = N(m, S + \gamma)$. While in general kernel density estimations works well only in low-dimensional spaces, this fits the bill for us, as we will only compute distances on single dimensional projections of the data.

**Cramer-Wold distance.** We are now ready to introduce the *Cramer-Wold distance*. In a nutshell, we propose to compute the squared distance between two samples by considering the mean squared $L_2$ distance between their smoothed projections over all single dimensional subspaces. By the squared $L_2$ distance between functions $f, g : \mathbb{R} \to \mathbb{R}$ we refer to $\|f - g\|_2^2 = \int |f(x) - g(x)|^2 dx$. A key feature of this distance is that it permits a closed-form in the case of normal distribution.

More precisely, the following algorithm fully defines the Cramer-Wold distance between two samples $X = (x_i)_{i=1..n}, Y = (y_j)_{j=1..k} \subset \mathbb{R}^D$ (for illustration of Steps 1 and 2 see Figure 1):

1. given $v$ in the unit sphere $S(0, 1) \subset \mathbb{R}^D$ consider the projections $v^T X = (v^T x_i)_{i=1..n}$ and $v^T Y = (v^T y_j)_{j=1..k}$,

2. compute the squared $L_2$ distance of the densities $\mathrm{sm}_\gamma(v^T X)$ and $\mathrm{sm}_\gamma(v^T X)$:
$$\|\mathrm{sm}_\gamma(v^T X) - \mathrm{sm}_\gamma(v^T Y)\|_2^2,$$

3. to obtain squared Cramer-Wold distance average (integrate) the above formula over all possible $v \in S_D$.

## 3 CRAMER-WOLD DISTANCE: THEORY

The key theoretical outcome of this paper is that the result of the computation of the Cramer-Wold distance from the previous section can be simplified to a closed form solution. Consequently, to compute the distance of two samples there is no need of finding the optimal transport like in WAE or the necessity to sample over the projections like in SWAE. For the case of simplicity we provide in this section the formulas for the distance between two samples and the distance of a sample from the standard normal density. The general definition of Cramer-Wold metric is presented in Appendix, Section A.

**Theorem 3.1.** *Let $X = (x_i)_{i=1..n}$, $Y = (y_j)_{j=1..n} \subset \mathbb{R}^D$ be given[2]. We formally define the squared Cramer-Wold distance by the formula*

$$d_{\mathrm{cw}}^2(X, Y) := \int_{S_D} \|\mathrm{sm}_\gamma(v^T X) - \mathrm{sm}_\gamma(v^T Y)\|_2^2 \, d\sigma_D(v).$$

*Then*

$$d_{\mathrm{cw}}^2(X, Y) = \frac{1}{2n^2\sqrt{\pi\gamma}} \Big( \sum_{ii'} \phi_D\big(\tfrac{\|x_i - x_{i'}\|^2}{4\gamma}\big) + \sum_{jj'} \phi_D\big(\tfrac{\|y_j - y_{j'}\|^2}{4\gamma}\big) - 2\sum_{ij} \phi_D\big(\tfrac{\|x_i - y_j\|^2}{4\gamma}\big) \Big), \quad (2)$$

*where $\phi_D(s) = {}_1F_1(\tfrac{1}{2}; \tfrac{D}{2}; -s)$ and ${}_1F_1$ is the Kummer's confluent hypergeometric function (see, e.g., Barnard et al. (1998)). Moreover, $\phi_D(s)$ has the following asymptotic formula valid for $D \geq 20$:*

$$\phi_D(s) \approx (1 + \tfrac{4s}{2D-3})^{-1/2}. \quad (3)$$

To prove the Theorem 3.1 we will need the following crucial technical proposition.

**Proposition 3.1.** *Let $z \in \mathbb{R}^D$ and $\gamma > 0$ be given. Then*

$$\int\limits_{S_D} N(v^T z, \gamma)(0) \, d\sigma_D(v) = \frac{1}{\sqrt{2\pi\gamma}} \phi_D\left(\frac{\|z\|^2}{2\gamma}\right). \quad (4)$$

*Proof.* By applying orthonormal change of coordinates without loss of generality we may assume that $z = (z_1, 0, \ldots, 0)$, and then $v^T z = z_1 v_1$ for $v = (v_1, \ldots, v_D)$. Consequently we get

$$\int_{S_D} N(v^T z, \gamma)(0) \, d\sigma_D(v) = \int_{S_D} N(z_1 v_1, \gamma)(0) \, d\sigma_D(v).$$

Making use of the formula for slice integration of functions on spheres (Axler et al., 1992, Corollary A.6) we get:

$$\int_{S_D} f \, d\sigma_D = \frac{V_{D-1}}{V_D} \int_{-1}^1 (1 - x^2)^{(D-3)/2} \int_{S_{D-1}} f(x, \sqrt{1-x^2}\,\zeta) \, d\sigma_{D-1}(\zeta) \, dx,$$

---

[2]For clarity of presentation we provide here the formula for the case of samples of equal size.

where $V_K$ denotes the surface volume of a sphere $S_K \subset \mathbb{R}^K$. Applying the above equality for the function $f(v_1, \ldots, v_D) = N(z_1 v_1, \gamma)(0)$ and $s = z_1^2/(2\gamma) = \|z\|^2/(2\gamma)$ we consequently get that the LHS of (4) simplifies to

$$\frac{V_{D-1}}{V_D} \frac{1}{\sqrt{2\pi\gamma}} \int_{-1}^1 (1-x^2)^{(D-3)/2} \exp(-sx^2)\, dx,$$

which completes the proof since $V_K = \frac{2 \cdot \pi^{\frac{K}{2}}}{\Gamma(\frac{K}{2})}$ and $\int_{-1}^1 \exp(-sx^2)(1-x^2)^{(D-3)/2}\, dx = \sqrt{\pi} \frac{\Gamma\left(\frac{D-1}{2}\right)}{\Gamma(\frac{D}{2})} \,_1F_1\left(\frac{1}{2}; \frac{D}{2}; -s\right)$. $\qquad\square$

*Proof of Theorem 3.1.* Directly from the definition of smoothing we obtain that

$$d_{\mathrm{cw}}^2(X, Y) = \int_{S_D} \left\| \tfrac{1}{n} \sum_i N(v^T x_i, \gamma) - \tfrac{1}{n} \sum_j N(v^T y_j, \gamma) \right\|_2^2 d\sigma_D(v). \tag{5}$$

Now applying the one-dimensional formula the the $L_2$-scalar product of two Gaussians:

$$\langle N(r_1, \gamma_1), N(r_2, \gamma_2)\rangle_2 = N(r_1 - r_2, \gamma_1 + \gamma_2)(0)$$

and the equality $\|f - g\|_2^2 = \langle f, f\rangle_2 + \langle g, g\rangle_2 - 2\langle f, g\rangle_2$ (where $\langle f, g\rangle_2 = \int f(x)g(x)dx$), we simplify the squared-$L_2$ norm in the integral of RHS of (5) to

$$\left\| \tfrac{1}{n} \sum_i N(v^T x_i, \gamma) - \tfrac{1}{n} \sum_i N(v^T y_j, \gamma) \right\|_2^2$$
$$= \tfrac{1}{n^2} \langle \sum_i N(v^T x_i, \gamma), \sum_i N(v^T x_i, \gamma)\rangle_2 + \tfrac{1}{n^2} \langle \sum_j N(v^T y_j, \gamma), \sum_j N(v^T y_j, \gamma)\rangle_2$$
$$- \tfrac{2}{n^2} \langle \sum_i N(v^T x_i, \gamma), \sum_j N(v^T y_j, \gamma)\rangle_2$$
$$= \tfrac{1}{n^2} \sum_{ii'} N(v^T(x_i - x_{i'}), 2\gamma)(0) + \tfrac{1}{n^2} \sum_{jj'} N(v^T(y_j - y_{j'}), 2\gamma)(0) - \tfrac{2}{n^2} \sum_{ij} N(v^T(x_i - y_j), 2\gamma)(0).$$

Applying directly Proposition 3.1 we obtain formula (2). Proof of the formula for the asymptotics of the function $\phi_D$ is provided in the Appendix. $\qquad\square$

Thus to estimate the distance of a given sample $X$ to some prior distribution $f$, one can follow the common approach and take the distance between $X$ and a sample from $f$. As the main theoretical result of the paper we view the following theorem, which says that in the case of standard Gaussian multivariate prior, we can completely reduce the need for sampling (we omit the proof since it is similar to that of Theorem 3.1).

**Theorem 3.2.** *Let $X = (x_i)_{i=1..n} \subset \mathbb{R}^D$ be a given sample. We formally define*

$$d_{\mathrm{cw}}^2(X, N(0, I)) := \int_{S_D} \|\mathrm{sm}_\gamma(v^T X) - \mathrm{sm}_\gamma(N(0, 1))\|_2^2 d\sigma_D(v).$$

*Then*

$$d_{\mathrm{cw}}^2(X, N(0, I)) = \frac{1}{2n^2\sqrt{\pi}} \left( \frac{1}{\sqrt{\gamma}} \sum_{i,j} \phi_D\left(\frac{\|x_i - x_j\|^2}{4\gamma}\right) + \frac{n^2}{\sqrt{1+\gamma}} - \frac{2n}{\sqrt{\gamma+\frac{1}{2}}} \sum_i \phi_D\left(\frac{\|x_i\|^2}{2+4\gamma}\right) \right). \tag{6}$$

One can easily obtain the general formula for the distance between mixtures of radial distributions. This follows from the fact that the Cramer-Wold distance is given by a scalar product $\langle \cdot, \cdot \rangle_{\mathrm{cw}}$ which has a closed-form for the product of two radial Gaussians:

$$\langle N(x, \alpha I), N(y, \beta I)\rangle_{\mathrm{cw}} = \frac{1}{\sqrt{2\pi(\alpha+\beta+2\gamma)}} \phi_D\left(\frac{\|x-y\|^2}{2(\alpha+\beta+2\gamma)}\right). \tag{7}$$

The above formula means that Cramer-Wold distance is defined by Cramer-Wold kernel, for more details see Appendix, Section A.

## 4 CRAMER-WOLD AUTOENCODER MODEL (CWAE)

This section is devoted to the construction of CWAE. Since we base our construction on the AutoEncoder, to establish notation let us formalize it here.

**AutoEncoder.** Let $X = (x_i)_{i=1..n} \subset \mathbb{R}^N$ be a given data set. The basic aim of AE is to transport the data to a typically, but not necessarily, less dimensional latent space $\mathcal{Z} = \mathbb{R}^D$ with reconstruction error as small as possible. Thus, we search for an encoder $\mathcal{E} : \mathbb{R}^n \to \mathcal{Z}$ and decoder $\mathcal{D} : \mathcal{Z} \to \mathbb{R}^n$ functions, which minimize the reconstruction error on the data set $X$:

$$MSE(X; \mathcal{E}, \mathcal{D}) = \frac{1}{n} \sum_{i=1}^{n} \|x_i - \mathcal{D}(\mathcal{E}x_i)\|^2.$$

**AutoEncoder based generative model.** CWAE, similarly to WAE, is a classical AutoEncoder model with modified cost function which forces the model to be generative, i.e. ensures that the data transported to the latent space comes from the (typically Gaussian) prior $f$. This statement is formalized by the following important remark, see also (Tolstikhin et al., 2017).

**Remark 4.1.** *Let* $\mathbf{X}$ *be an $N$-dimensional random vector from which our data set was drawn, and let* $\mathbf{Y}$ *be a random vector with a density $f$ on latent $\mathcal{Z}$.*

*Suppose that we have constructed functions $\mathcal{E} : \mathbb{R}^N \to \mathcal{Z}$ and $\mathcal{D} : \mathcal{Z} \to \mathbb{R}^N$ (representing the encoder and the decoder) such that[3]*

1. *$\mathcal{D}(\mathcal{E}x) = x$ for $x \in \text{image}(\mathbf{X})$,*

2. *random vector $\mathcal{E}\mathbf{X}$ has the distribution $f$.*

*Then by the point 1 we obtain that $\mathcal{D}(\mathcal{E}\mathbf{X}) = \mathbf{X}$, and therefore*

$$\mathcal{D}\mathbf{Y} \text{ has the same distribution as } \mathcal{D}(\mathcal{E}\mathbf{X}) = \mathbf{X}.$$

*This means that to produce samples from $\mathbf{X}$ we can instead produce samples from $\mathbf{Y}$ and map them by the decoder $\mathcal{D}$.*

Since an estimator of the image of the random vector $\mathbf{X}$ is given by its sample $X$, we conclude that a generative model is correct if it has small reconstruction error and resembles the prior distribution in the latent. Thus, to construct a generative AutoEncoder model (with Gaussian prior), we add to its cost function a measure of distance of a given sample from normal distribution.

**CWAE cost function.** Once the crucial ingredient of CWAE is ready, we can describe its cost function. To ensure that the data transported to the latent space $\mathcal{Z}$ are distributed according to the standard normal density, we add to the cost function logarithm[4] of the Cramer-Wold distance from standard multivariate normal density $d_{\text{cw}}^2(X, N(0, I))$:

$$\text{cost}(X; \mathcal{E}, D) = \log d_{\text{cw}}^2(\mathcal{E}X, N(0, I)) + MSE(X; \mathcal{E}, \mathcal{D}). \tag{8}$$

Since the use of special functions involved in the formula for Cramer-Wold distance might be cumbersome, we apply in all experiments (except for the illustrative 2D case) the asymptotic form (12) of function $\phi_D$:

$$2\sqrt{\pi} d_{\text{cw}}^2(X) \approx \frac{1}{n^2} \sum_{ij} (\gamma_n + \frac{\|x_i - x_j\|^2}{2D-3})^{-1/2} + (1 + \gamma_n)^{-1/2} - \frac{2}{n} \sum_i (\gamma_n + \frac{1}{2} + \frac{\|x_i\|^2}{2D-3})^{-1/2},$$

where $\gamma_n = (\frac{4}{3n})^{2/5}$ is chosen by the Silverman's rule of thumb (Silverman, 1986).

**Comparison with WAE and SWAE models.** Finally, let us briefly recapitulate differences between the introduced CWAE, WAE variants of (Tolstikhin et al., 2017) and SWAE (Kolouri et al., 2018). In contrast to WAE-MMD and SWAE, CWAE model *does not* require sampling from normal distribution (as in WAE-MMD) or over slices (as in SWAE) to evaluate its cost function, and in this sense uses a closed formula cost function. In contrast to WAE-GAN, our objective does not require a separately trained neural network to approximate the optimal transport function, thus avoiding pitfalls of adversarial training. In this paper we are interested in WAE-MMD and SWAE models, which *do not use parameterized distance functions, e.g. trained adversarially like in WAE-GAN*. However, in future work we plan to introduce an adversarial version of CWAE and compare it with WAE-GAN.

---

[3]We recall that for function (or in particular random vector) $\mathbf{X} : \Omega \to \mathbb{R}^D$ by $\text{image}(\mathbf{X})$ we denote the set consisting of all possible values $\mathbf{X}$ can attain, i.e. $\{\mathbf{X}(\omega) : \omega \in \Omega\}$.

[4]We take the logarithm of the Cramer-Wold distance to improve balance between the two terms in the objective function.

## 5 EXPERIMENTS

In this section we empirically validate the proposed CWAE model on standard benchmarks for generative models: CelebA, Cifar-10 and MNIST. We will compare CWAE model with WAE-MMD (Tolstikhin et al., 2017) and SWAE (Kolouri et al., 2018). As we will see, our results match those of WAE-MMD, and in some cases improve upon SWAE, while using a simpler to optimize cost function (see the previous section for a more detailed discussion). The rest of this section is structured as follows. In Section 5.2 we report results on standard qualitative tests, as well as a visual investigations of the latent space. In Section 5.3 we will turn our attention to quantitative tests using Fréchet Inception Distance and other metrics.

### 5.1 EXPERIMENTATION SETUP

In the experiment we have used two basic architecture types. Experiments on MNIST were performed using a feedforward network for both encoder and decoder, and a 20 neuron latent layer, all with ReLU activations. In case of CIFAR-10 and CelebA data sets we used convolution-deconvolution architectures. Please refer to Appendix E for full details.

### 5.2 QUALITATIVE TESTS

The quality of a generative model is typically evaluated by examining samples or interpolations. We present such a comparison between CWAE with WAE-MMD in Figure 2. We follow the same procedure as in (Tolstikhin et al., 2017). In particular, we use the same base neural architecture for both CWAE and WAE-MMD. We consider for each model (i) interpolation between two random examples from the test set (leftmost in Figure 2), (ii) reconstruction of a random example from the test set (middle column in Figure 2), and finally a sample reconstructed from a random point sampled from the prior distribution (right column in Figure 2). The experiment shows that *there are no perceptual differences between CWAE and WAE-MMD generative distribution.*

Table 1: CWAE achieves similar FID (lower is better) and sharpness (higher is better) to WAE-MMD on the original WAE architecture (see Appendix E for details).

| Algorithm | FID | Sharpness |
|---|---|---|
| SWAE | 72 | 0.008 |
| VAE | 63 | 0.003 |
| WAE-MMD | 55 | 0.006 |
| CWAE | 54 | 0.006 |
| WAE-GAN | 42 | 0.006 |
| True data | 2 | 0.020 |

In the next experiment we qualitatively assess normality of the latent space. This will allow us to ensure that CWAE does not compromise on the normality of its latent distribution, which recall is part of the cost function for all the models except AE. We compare CWAE[5] with AE, VAE, WAE and SWAE on the MNIST data with using 2-dimensional latent space and a two dimensional Gaussian prior distribution. Results are reported in Figure 3. As is readily visible, the latent distribution of CWAE is as close, or perhaps even closer, to the normal distribution than that of the other models. Furthermore, the AutoEncoder presented in the second figure is noticeably different from a Gaussian distribution, which is to be expected because it does not optimize for normality in contrast to the other models.

To summarize, both in terms of perceptual quality and satisfying normality objective, CWAE matches WAE-MMD. The next section will provide more quantitative studies.

### 5.3 QUANTITATIVE TESTS

In order to quantitatively compare CWAE with other models, in the first experiment we follow the common methodology and use the Fréchet Inception Distance (FID) introduced by Heusel et al. (2017). Further, we evaluate the sharpness of generated samples using the Laplace filter following Tolstikhin et al. (2017). Results for CWAE and WAE are summarized in Tab. 1. In agreement with the qualitative studies, we observe FID and sharpness scores of CWAE to be similar to WAE-MMD.

---

[5]Since (3) is valid for dimensions $D \geq 20$, to implement CWAE in 2-dimensional latent space we apply equality $_1F_1(1/2, 1, -s) = e^{-\frac{s}{2}} I_0\left(\frac{s}{2}\right)$ jointly with the approximate formula (Abramowitz & Stegun, 1964, page 378) for the Bessel function of the first kind $I_0$, for more details see Appendix C.

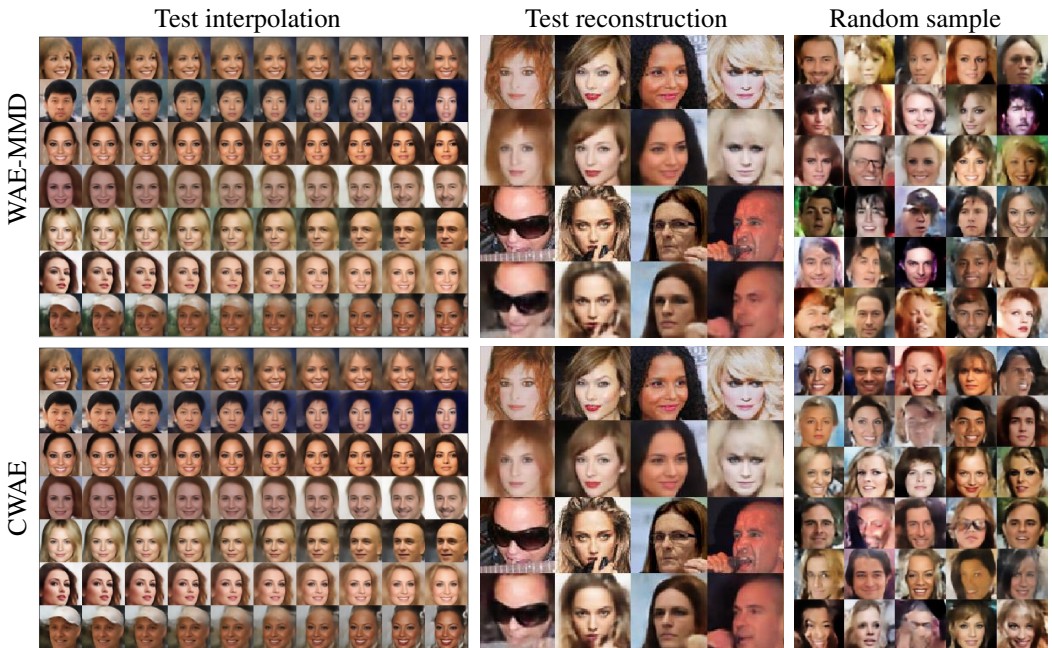

Figure 2: CWAE achieves perceptually similar results to WAE-MMD. Results of WAE-MMD (first row) and CWAE (second row) on CelebA data set of original WAE-MMD architecture. **Left:** Interpolations between two examples from the test distribution (left to right, in each row). **Middle:** Reconstruction of examples from the test distribution; odd rows correspond to the real test points. **Right:** Reconstructed examples from a random samples from the prior distribution.

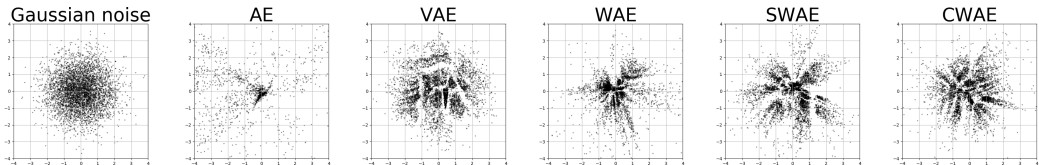

Figure 3: Latent distribution of CWAE is close to the normal distribution. Each subfigure presents points sampled from two-dimensional latent spaces, AE, VAE, WAE, SWAE, and CWAE (left to right). All trained on the MNIST data set.

Next, by comparing training time between CWAE and other models, we found that for batch-sizes up to $1024$, which covers the range of batch-sizes used typically for training autoencoders, CWAE is faster (in terms of time spent per batch) than other models. More precisely, CWAE is approximately $2\times$ faster up to $256$ batch-size. Details are relegated to the Appendix D.

Finally, motivated by Remark 4.1 we propose a novel method for quantitative assessment of the models based on their comparison to standard normal distribution in the latent. To achieve this we have decided to use one of the most popular statistical normality tests, i.e. Mardia tests (Henze, 2002). Mardia's normality tests are based on verifying whether the skewness $b_{1,D}(\cdot)$ and kurtosis $b_{2,D}(\cdot)$ of a sample $X = (x_i)_{i=1..n} \subset \mathbb{R}^D$:

$$b_{1,D}(X) = \tfrac{1}{n^2} \sum_{j,k} (x_j^T x_k)^3, \text{ and } b_{2,D}(X) = \tfrac{1}{n} \sum_j \|x_j\|^4,$$

are close to that of standard normal density. The expected Mardia's skewness and kurtosis for standard multivariate normal distribution is $0$ and $D(D+2)$, respectively. To enable easier comparison in experiments we consider also the value of the normalized Mardia's kurtosis given by $b_{2,D}(X) - D(D+2)$, which equals zero for the standard normal density.

Table 2: Comparison between different models output distributions and the normal distribution, together with reconstruction error. All models outputs except AE are similarly close to the normal distribution. Normality is assessed by comparing Mardia's skewness, kurtosis, and the reconstruction error. For reference FID scores are provided as well (except for MNIST, where it is not defined).

| Data set | Method | AE | VAE | WAE | SWAE | CWAE |
|----------|--------|-----:|-----:|-----:|-----:|-----:|
| MNIST | Skewness | 659.67 | 0.49 | 82.12 | 55.59 | 52.83 |
| | Kurtosis (normalized) | 749.58 | -410.69 | 35.61 | -37.43 | 95.77 |
| | Reconstruction error | 2.10 | 4.12 | 2.11 | 2.11 | 2.13 |
| CIFAR10 | Skewness | 11444.35 | 3.07 | 1893.10 | 996.01 | 171.67 |
| | Kurtosis (normalized) | -2219.50 | -4158.33 | 2346.49 | 193.15 | 1943.35 |
| | Reconstruction error | 49.67 | 82.82 | 45.80 | 44.84 | 45.52 |
| | *FID score error* | 400.14 | 218.43 | 146.34 | 145.04 | 121.16 |
| CelebA | Skewness | 59770025.50 | 22.07 | 301.71 | 196.64 | 59.22 |
| | Kurtosis (normalized) | 1363931.65 | 53.09 | 942.68 | 507.39 | 307.29 |
| | Reconstruction error | 139.30 | 142.46 | 139.10 | 138.23 | 138.54 |
| | *FID score error* | 307.70 | 95.35 | 96.30 | 100.56 | 97.22 |

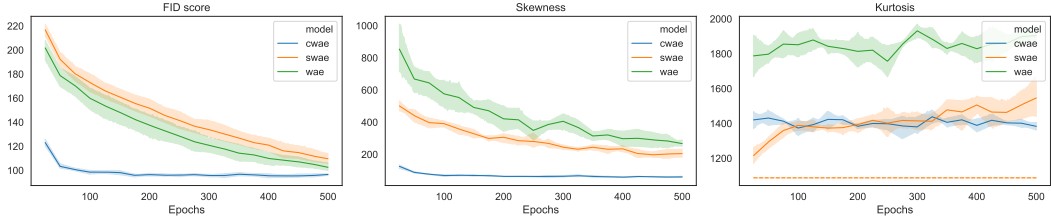

Figure 4: Metrics assessing normality of the model output distributions, during training: FID score, Mardia's skewness and kurtosis of models WAE, SWAE and CWAE, on the CelebA test set. The values given are mean from 5 experiments of differently initialised models together with standard deviation confidence intervals. Low standard deviation of CWAE indicates its stability during learning. Optimal value of kurtosis, (i.e. for normal distribution) is given by a dash line.

Results are presented in Figure 4 and Table 2. In Figure 4 we report for CelebA data set the value of FID score, Mardia's skewness and kurtosis during learning process of AE, VAE, WAE, SWAE and CWAE (measured on the validation data set). WAE, SWAE and CWAE models obtain the best reconstruction error, comparable to AE. VAE model exhibits a sightly worse reconstruction error, but values of kurtosis and skewness indicating their output is closer to normal distribution. As expected, the output of AE is far from normal distribution; its kurtosis and skewness grow during learning. This arguably less standard evaluation, which we hope will find adoption in the community, serves as yet another evidence that *CWAE has strong generative capabilities which at least match performance of WAE-MMD*. Moreover we observe that VAE model's output distribution is closest to the normal distribution, at the expense of the reconstruction error, which is reflected by the blurred reconstructions typically associated with VAE model.

Moreover, motivated by the above approach based on normality tests[6] we have verified how the Cramer-Wold metric works as a Gaussian goodness of fit, however, the results were not satisfactory. The tests based on Cramer-Wold metric were, in general, in the middle of compared tests (Mardia, Henze-Zirkler and Royston tests).

On the whole, WAE-MMD and CWAE achieve, practically speaking, the same level of performance in terms of FID score, sharpness, and our newly introduced normality test. Additionally, CWAE fares better in many of these metrics than SWAE.

---

[6]For more information on the statistical tests based on the kernel approach we refer the reader to (Muandet et al., 2017, Subsection 3.5).

## 6 CONCLUSIONS

In the paper we have presented a new autoencoder based generative model CWAE, which matches results of WAE-MMD, while using *a cost function given by a simple closed analytic formula.* We hope this result will encourage future work in developing simpler to optimize analogs of strong neural models.

Crucial in the construction of CWAE is the use of the developed Cramer-Wold metric between samples and distributions, which can be effectively computed for Gaussian mixtures. As a consequence we obtain a reliable measure of the divergence from normality. Future work could explore use of the Cramer-Wold distance in other settings, in particular in adversarial models.

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

APPENDICES

## A CRAMER-WOLD KERNEL

In this section we first formally define the Cramer-Wold metric, and later show that it is given by a characteristic kernel which has closed-form for spherical Gaussians. For more information on the kernels, and in general kernel embedding of distributions we refer the reader to (Muandet et al., 2017).

Let us first introduce the general definition of the cw-metric. To do so we generalize the notion of smoothing for arbitrary measures $\mu$ by the formula:

$$\text{sm}_\gamma(\mu) = \mu * N(0, \gamma I),$$

where $*$ denotes the convolution operator for two measures, and we identify the normal density $N(0, \gamma I)$ with the measure it introduces. It is well-known that the resulting measure has the density given by

$$x \to \int N(x, \gamma I)(y) d\mu(y).$$

Clearly

$$\text{sm}_\gamma(N(0, \alpha I)) = N(0, (\alpha + \gamma)I)).$$

Moreover, by applying the characteristic function one obtains that if the smoothing of two measures coincide, then the measures also coincide:

$$\text{sm}_\gamma(\mu) = \text{sm}_\gamma(\mu) \implies \mu = \nu. \tag{9}$$

We also need to define the transport of the density by the projection $x \to v^T x$, where $v$ is chosen from the unit sphere $S_D$. The definition is formulated so that if $\mathbf{X}$ is a random vector with density $f$, then $f_v$ is the density of the random vector $\mathbf{X}_v := v^T \mathbf{X}$. Then we have

$$f_v(r) = \int_{y:y-rv\perp v} f(z) d_{D-1}(z),$$

where $d_{D-1}$ denotes the $D-1$-dimensional Lebesgue measure. In general, if $\mu$ is a measure on $\mathbb{R}^D$, then $\mu_v$ is the measure defined on $\mathbb{R}$ by the formula

$$\mu_v(A) = \mu(\{x : v^T x \in A\}).$$

Since, if a random vector $\mathbf{X}$ has the density $N(a, \gamma I)$, then the random variable $\mathbf{X}_v$ has the density $N(v^T a, \alpha)$, we directly conclude that

$$N(a, \gamma I)_v = N(v^T a, \gamma).$$

It is also worth noticing, that due to the fact that the projection of a Gaussian is a Gaussian, the smoothing and projection operators commute, i.e.:

$$\text{sm}_\gamma(\mu_v) = (\text{sm}_\gamma \mu)_v.$$

Given fixed $\gamma > 0$, the two above notions allow us to formally define the cw-distance of two measures $\mu$ and $\nu$ by the formula

$$d_{\text{cw}}^2(\mu, \nu) = \int_{S_D} \|\text{sm}_\gamma(\mu_v) - \text{sm}_\gamma(\nu_v)\|_{L_2}^2 d\sigma_D(v). \tag{10}$$

First observe that this implies that cw-distance is given by the kernel function

$$k(\mu, \nu) = \int_{S_D} \langle \text{sm}_\gamma(\mu_v), \text{sm}_\gamma(\nu_v) \rangle_{L_2} d\sigma_D(v).$$

Let us now prove that the function $d_{\text{cw}}$ defined by equation 10 is a metric (which, in the kernel function literature means that the kernel is characteristic).

**Theorem A.1.** *The function $d_{\text{cw}}$ is a metric.*

*Proof.* Since $d_{\mathrm{cw}}$ comes from a scalar product, we only need to show that if a distance of two measures is zero, the measures coincide.

So let $\mu, \nu$ be given measures such that $d_{\mathrm{cw}}(\mu, \nu) = 0$. This implies that

$$\mathrm{sm}_\gamma(\mu_v) = \mathrm{sm}_\gamma(\nu_v).$$

By equation 9 this implies that $\mu_v = \nu_v$. Since this holds for all $v \in S_D$, by the Cramer-Wold Theorem we obtain that $\mu = \nu$. $\qquad\square$

Thus we can summarize the above by saying that the Cramer-Wold kernel is a characteristic kernel which has the closed-form the scalar product of two radial Gaussians given by equation 7:

$$\langle N(x, \alpha I), N(y, \beta I) \rangle_{\mathrm{cw}} = \frac{1}{\sqrt{2\pi(\alpha+\beta+2\gamma)}} \phi_D \left( \frac{\|x-y\|^2}{2(\alpha+\beta+2\gamma)} \right).$$

**Remark A.1.** *Observe, that except for the Gaussian kernel it is the only kernel which has the closed form for the spherical Gaussians, which as we discuss in the next section is important, as the RBF (Gaussian) kernels cannot be successfully applied in AutoEncoder based generative models. The reason is that the derivative of Gaussian decreases to fast, and therefore it does not enforce the proper learning of the model, see also the comments in (Tolstikhin et al., 2017, Section 4, WAE-GAN and WAE-MMD specifics).*

## B   CWAE VS WAE-MMD

In this section we are going to compare CWAE model to WAE-MMD. In particular we show that CWAE can be seen as the intersection of the sliced-approach together with MMD-based models.

Since both WAE and CWAE use kernels to discriminate between sample and normal density, to compare the models we first describe the WAE model.

WAE cost function for a given characteristic kernel $k$ and sample $X = (x_i)_{i=1..n} \subset \mathbb{R}^D$ (in the $D$-dimensional latent) is given by

$$\mathrm{WAE\ cost} = MSE + \lambda \cdot d_k^2(X, Y),$$

where $Y = (y_i)_{i=1..n}$ is a sample from the standard normal density $N(0, I)$, and $d_k^2(X, Y)$ denotes the kernel-based distance between the probability distributions representing $X$ and $Y$, that is $\frac{1}{n} \sum_i \delta_{x_i}$ and $\frac{1}{n} \sum_i \delta_{y_i}$, where $\delta_z$ denotes the atom Dirac measure at $z \in \mathbb{R}^D$. The inverse multiquadratic kernel $k$ is chosen as default

$$k(x, y) = \frac{C}{C + \|x - y\|_2^2},$$

where in experiments in (Tolstikhin et al., 2017) a value $C = 2D\sigma^2$ was used, where $\sigma$ is the hyper-parameter denoting the size of the normal density. Thus the model has hyper-parameters $\lambda$ and $\sigma$, which were chosen to be $\lambda = 10, \sigma^2 = 1$ in MNIST, $\lambda = 100, \sigma^2 = 2$ in CelebA. Observe that the hyper-parameters do not depend on the sample size and that, in general, the WAE-MMD model hyper-parameters have to be chosen by hand.

Now let us describe the CWAE model. CWAE cost function for a sample $X = (x_i)_{i=1..n} \subset \mathbb{R}^D$ (in the $D$-dimensional latent) is given by

$$\mathrm{CWAE\ cost} = MSE + \log d_{\mathrm{cw}}^2(X, N(0, I)),$$

where distance between the sample and standard normal distribution is taken with respect to the Cramer-Wold kernel with a regularizing hyperparameter $\gamma$ given by the Silverman's rule of thumb (the motivation for such a choice of hyper-parameters is explained in Section 2).

Thus, we have the following differences:

- Due to the properties of Cramer-Wold kernel, in the distance we are able to substitute the sample estimation of $d_k^2(X, N(0, I))$ given in WAE-MMD by $d_{\mathrm{cw}}^2(X, Y)$ by its exact formula.
- CWAE, as compared to WAE, has no hyper-parameters:

1. In our preliminary experiments we have observed that in many situations (like in the case of log-likelihood), taking the logarithm of the nonnegative factors of the cost function, which we aim to minimize to zero, improves the learning process. Motivated by this, instead of taking the additional weighting hyper-parameter $\lambda$ (as in WAE-MMD), whose aim is to balance the MSE and divergence terms, we take the logarithm of the divergence. Automatically (independently of dimension) balance those terms in the learning process.

2. The choice of regularization hyper-parameter is given by the Silverman's rule of thumb, and depends on the sample size (contrary to WAE-MMD, where the hyper-parameters are chosen by hand, and in general do not depend on the sample size).

Summarizing, in CWAE model, contrary to WAE-MMD, we do not have to choose hyper-parameters. Moreover, since we do not have the noise in the learning process given by the random choice of the sample $Y$ from $N(0, I)$, the learning should be more stable. As a consequence, see Figure 7, CWAE in generally learns faster then WAE-MMD, and has smaller standard deviation of the cost-function during the learning process.

## C  COMPUTATION OF $\phi_D$

In this section we consider the estimation of values of the function

$$\phi_D(s) = {}_1F_1(\tfrac{1}{2}; \tfrac{D}{2}; -s) \text{ for } s \geq 0,$$

which is crucial in the formulation for the Cramer-Wold distance. First we will provide its approximate asymptotic formula valid for dimensions $D \geq 20$, and then we shall consider the special case of $D = 2$ (see Figure 5)

To do so, let us first recall (Abramowitz & Stegun, 1964, Chapter 13) that the Kummer's confluent hypergeometric function ${}_1F_1$ (denoted also by $M$) has the following integral representation

$${}_1F_1(a, b, z) = \frac{\Gamma(b)}{\Gamma(a)\Gamma(b - a)} \int_0^1 e^{zu} u^{a-1}(1 - u)^{b-a-1} \, du,$$

valid for $a, b > 0$ such that $b > a$. Since we consider that latent is at least of dimension $D \geq 2$, it follows that

$$\phi_D(s) = \frac{\Gamma(\frac{D}{2})}{\Gamma(\frac{1}{2})\Gamma(\frac{D}{2} - \frac{1}{2})} \int_0^1 e^{-su} u^{-1/2}(1 - u)^{D/2-3/2} \, du.$$

By making a substitution $u = x^2$, $du = 2x dx$, we consequently get

$$\phi_D(s) = 2 \cdot \frac{\Gamma(D/2)}{\Gamma(1/2)\Gamma(D/2-1/2)} \int_0^1 e^{-sx^2}(1 - x^2)^{(D-3)/2} \, dx$$

$$= \frac{\Gamma(D/2)}{\Gamma(1/2)\Gamma(D/2-1/2)} \int_{-1}^1 e^{-sx^2}(1 - x^2)^{(D-3)/2} \, dx. \tag{11}$$

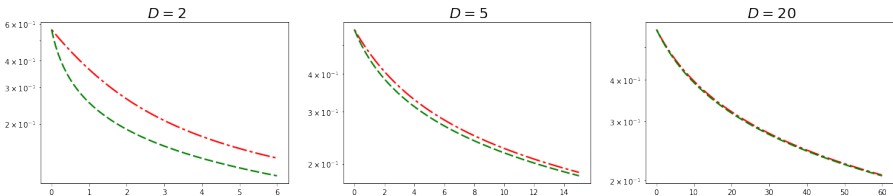

Figure 5: Comparison of $\phi_D$ value (red line) with the approximation given by equation 12 (green line) in the case of dimensions $D = 2, 5, 20$. Observe that for $D = 20$, the functions practically coincide.

**Proposition C.1.** *For large[7] $D$ we have*

$$\phi_D(s) \approx (1 + \tfrac{4s}{2D-3})^{-1/2} \text{ for all } s \geq 0. \tag{12}$$

*Proof.* By (11) we have to estimate asymptotics of

$$\phi_D(s) = \frac{\Gamma(\frac{D}{2})}{\Gamma(\frac{1}{2})\Gamma(\frac{D}{2} - \frac{1}{2})} \int_{-1}^{1} e^{-sx^2}(1 - x^2)^{(D-3)/2}\, dx.$$

Since for large $D$, for all $x \in [-1, 1]$ we have

$$(1 - x^2)^{(D-3)/2}e^{-sx^2} \approx (1 - x^2)^{(D-3)/2} \cdot (1 - x^2)^s = (1 - x^2)^{s+(D-3)/2},$$

we get

$$\phi_D(s) \approx \frac{\Gamma(\frac{D}{2})}{\Gamma(\frac{D-1}{2})\sqrt{\pi}} \cdot \int_{-1}^{1} (1 - x^2)^{s+(D-3)/2}\, dx = \frac{\Gamma(\frac{D}{2})}{\Gamma(\frac{D-1}{2})\sqrt{\pi}} \cdot \sqrt{\pi}\frac{\Gamma(s + \frac{D}{2} - \frac{1}{2})}{\Gamma(s + \frac{D}{2})}.$$

To simplify the above we apply the formula (1) from (Tricomi & Erdélyi, 1951):

$$\frac{\Gamma(z + \alpha)}{\Gamma(z + \beta)} = z^{\alpha-\beta}(1 + \frac{(\alpha - \beta)(\alpha + \beta - 1)}{2z} + O(|z|^{-2})),$$

with $\alpha, \beta$ fixed so that $\alpha + \beta = 1$ (so only the error term of order $O(|z|^{-2})$ remains), and get

$$\frac{\Gamma(\frac{D}{2})}{\Gamma(\frac{D-1}{2})} = \frac{\Gamma((\frac{D}{2} - \frac{3}{4}) + \frac{3}{4})}{\Gamma((\frac{D}{2} - \frac{3}{4}) + \frac{1}{4})} \approx \left(\frac{D}{2} - \frac{3}{4}\right)^{\frac{1}{2}} \text{ and } \frac{\Gamma(s + \frac{D}{2} - \frac{1}{2})}{\Gamma(s + \frac{D}{2})} \approx \left(s + \frac{D}{2} - \frac{3}{4}\right)^{-\frac{1}{2}}. \tag{13}$$

Summarizing,

$$\phi_D(s) \approx \frac{(\frac{D}{2} - \frac{3}{4})^{1/2}}{(s + \frac{D}{2} - \frac{3}{4})^{1/2}} = (1 + \tfrac{4s}{2D-3})^{-1/2}.$$

$\square$

In general one can obtain the iterative direct formulas for function $\phi_D$ with the use of $\mathrm{erf}$ and modified Bessel functions of the first kind $I_0$ and $I_1$, but for large $D$ they are of little numerical value. We consider here only the special case $D = 2$ since it is used in the paper for illustrative reasons in the latent for the MNIST data set. Since we have the equality (Gradshteyn & Ryzhik, 2015, (8.406.3) and (9.215.3)):

$$\phi_2(s) = {}_1F_1(\tfrac{1}{2}, 1, -s) = e^{-\frac{s}{2}}I_0\left(\frac{s}{2}\right),$$

to practically implement $\phi_2$ we apply the approximation of $I_0$ from (Abramowitz & Stegun, 1964, page 378) given in the following remark.

**Remark C.1.** *Let $s \geq 0$ be arbitrary and let $t = s/7.5$. Then*

$\phi_2(s) \approx$

$$\begin{cases} e^{-\frac{s}{2}} \cdot (1 + 3.5156229t^2 + 3.0899424t^4 + 1.2067492t^6 + .2659732t^8 + .0360768t^{10} + .0045813t^{12}) \\ \hspace{10cm} \text{for } s \in [0, 7.5], \\[2mm] \sqrt{\frac{2}{s}} \cdot (.39894228 + .01328592t^{-1} + .00225319t^{-2} - .00157565t^{-3} + .0091628t^{-4} - .02057706t^{-5} \\ \hspace{4cm} + .02635537t^{-6} - .01647633t^{-7} + .00392377t^{-8}) \\ \hspace{10cm} \text{for } s \geq 7.5. \end{cases}$$

## D  COMPARISON OF LEARNING TIMES

Figure 6 gives comparison of mean learning time for different most frequently used batch-sizes. Time spent on processing a batch is actually smaller for CWAE for a practical range of batch-sizes $[32, 512]$. For batch-sizes larger than 1024, CWAE is slower due to its quadratic complexity with respect to the batch-size. However, we note that batch-sizes larger even than 512 are relatively rarely used in practice for training autoencoders.

---

[7]In practice we can take $D \geq 20$.

Figure 6: Comparison of mean batch learning time (times are in log-scale) for different algorithms in seconds, all for the same architecture like the one in Tolstikhin et al. (2017) and all requiring similar number of epochs to train the full model. This times may differ for computer architectures with more/less memory on a GPU card.

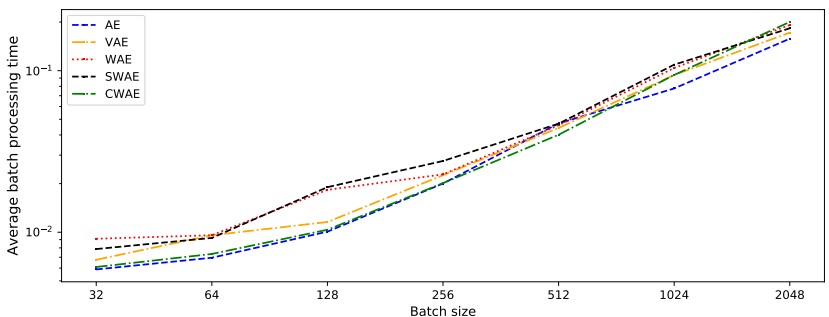

# E  ARCHITECTURE DETAILS

MNIST ($28 \times 28$ images) an encoder-decoder feedforward architecture:

**encoder**  three feed-forward ReLU layers, 200 neurons each

   **latent**  20-dimensional,

**decoder**  three feed-forward ReLU layers, 200 neurons each.

CelebA (with images centered and cropped to $64 \times 64$ with 3 color layers) a convolution-deconvolution network:

**encoder**  four convolution layers with $4 \times 4$ filters and $2 \times 2$ strides (consecutively 32, 32, 64, and 64 output channels), all ReLU activations,

two dense layers (1024 and 256 ReLU neurons)

   **latent**  64-dimensional,

**decoder**  first two dense layers (256 and 1024 ReLU neurons),

three transposed-convolution layers with $4 \times 4$ filters with $2 \times 2$ strides (consecutively 64, 32, 32 channels) with ReLU activation,

transposed-convolution $4 \times 4$ with $2 \times 2$ stride, 3 channels, and sigmoid activation.

CIFAR-10 dataset ($32 \times$ images with three color layers): a convolution-deconvolution network

**encoder**  four convolution layers with $2 \times 2$ filters, the second one with $2 \times 2$ strides, other non-strided (3, 32, 32, and 32 channels) with ReLU activation,

128 ReLU neurons dense layer,

   **latent**  with 64 neurons,

**decoder**  two dense $ReLU$ layers with 128 and 8192 neurons,

two transposed-convolution layers with $2 \times 2$ filters (32 and 32 channels) and ReLU activation,

a transposed convolution layer with $3 \times 3$ filter and $2 \times 2$ strides (32 channels) and ReLU activation,

a transposed convolution layer with $2 \times 2$ filter (3 channels) and sigmoid activation.

The last layer returns the reconstructed image. The results for all above architectures are given in Table 2. All networks were trained with the Adam optimizer (Kingma & Ba, 2014). The hyper-parameters used were $learning\ rate = 0.001$, $\beta1 = 0.9$, $\beta2 = 0.999$, $\epsilon = 1e - 8$. MNIST models were trained for 500 epochs, both CIFAR-10 and CelebA for 200.

Additionally, to have a direct comparison to WAE-MMD model on CelebA, an identical architecture was used as that in Tolstikhin et al. (2017) utilized for the WAE-MMD model (WAE-GAN architecture is, naturally, different):

**encoder**      four convolution layers with $5 \times 5$ filters, each layer followed by a batch normalization (consecutively 128, 256, 512, and 1024 channels) and ReLU activation,

**latent**      64-dimensional,

**decoder**      dense 1024 neuron layer,

three transposed-convolution layers with $5 \times 5$ filters, and each layer followed by a batch normalization with ReLU activation (consecutively 512, 256, and 128 channels),

transposed-convolution layer with $5 \times 5$ filter and 3 channels, clipped output value.

The results for this architecture for CWAE compared to VAE and WAE-MMD models are given in Table 1.

Similarly to Tolstikhin et al. (2017), models were trained using Adam with for 55 epochs, with the same optimizer parameters.

## F  TRAINING RESULTS FOR THE CELEBA DATASET

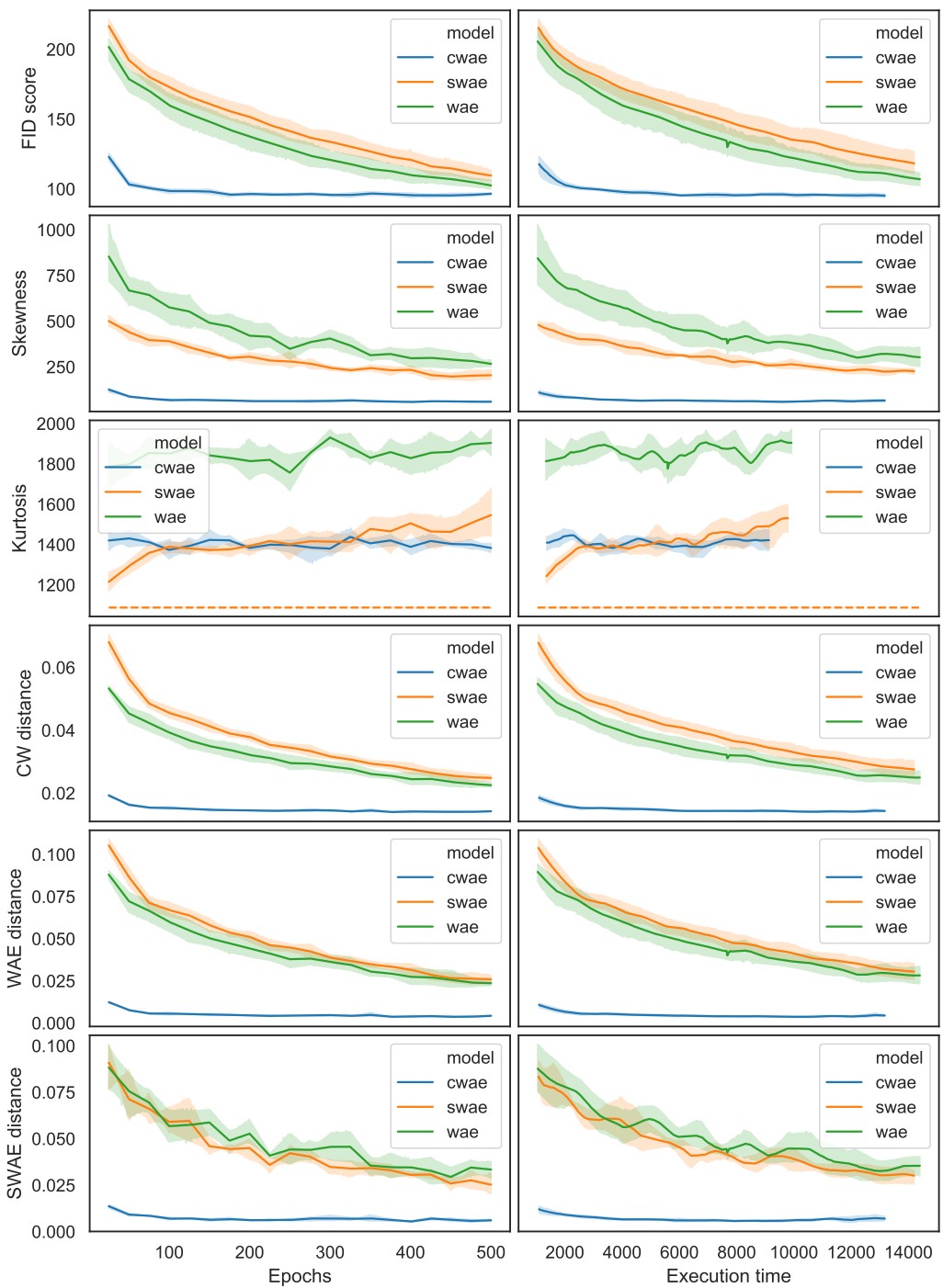

Figure 7: CelebA trained CWAE, WAE, and SWAE models with FID score, kurtosis and skewness, as well as CW-, WAE-, and SWAE-distances. Results are given both related to epochs and learning time. All values are mean from 5 models trained for each architecture. Confidence intervals represent the standard deviation. Optimum kurtosis is marked by a dash line.

## G    TRAINING RESULTS FOR THE CIFAR10 DATASET

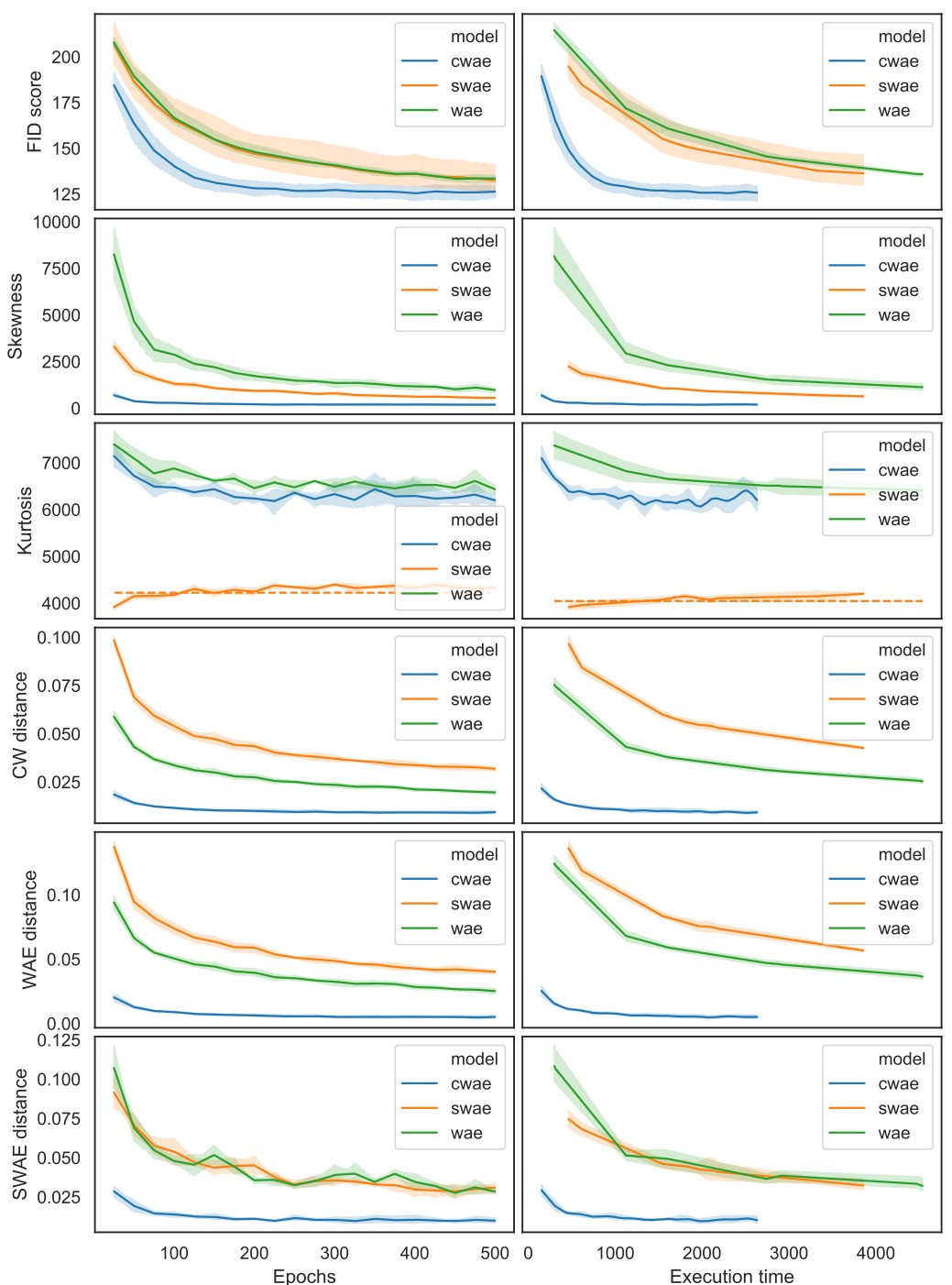

Figure 8: CIFAR10 trained CWAE, WAE, and SWAE models with FID score, kurtosis and skewness, as well as CW-, WAE-, and SWAE-distances. Results are given both related to epochs and learning time. All values are mean from 5 models trained for each architecture. Confidence intervals represent the standard deviation. Optimum kurtosis is marked by a dash line.

# H  ADDITIONAL FIGURES

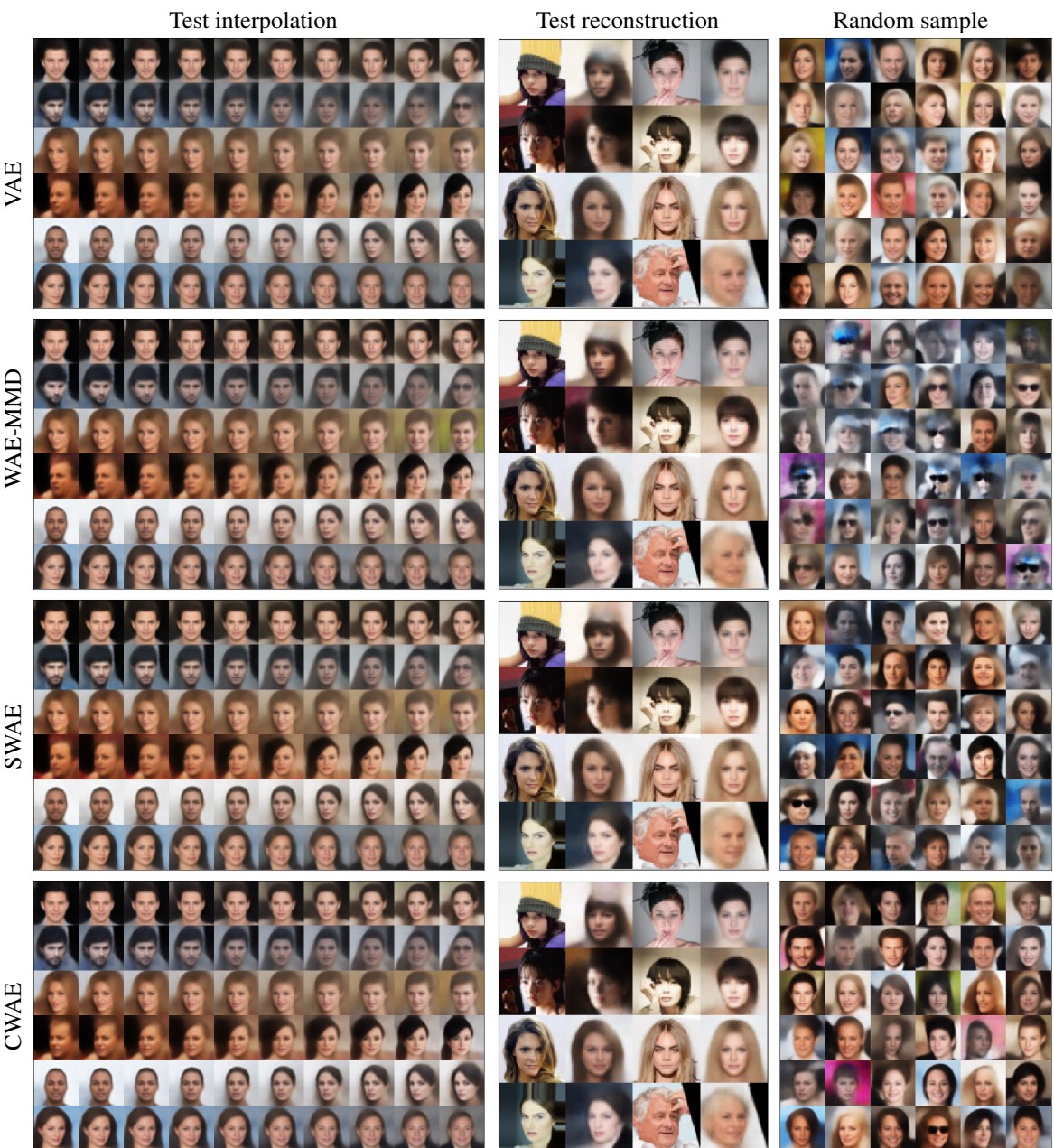

Figure 9: Results of VAE, WAE-MMD, SWAE, and CWAE models trained on CelebA dataset using the WAE architecture from Tolstikhin et al. (2017). In "test reconstructions" odd rows correspond to the real test points.

