# OpenReview forum: "Cramer-Wold AutoEncoder"
_ICLR.cc/2019/Conference_

### Official Review · AnonReviewer2 · 2018-11-03
**Nice idea & paper, but needs to highlight at least one practical advantage**

**Rating:** 6
**Confidence:** 4

**Review:**

This paper proposes a WAE variant based on a new statistical distance between the encoded data distribution and the latent prior distribution that can be computed in closed form without drawing samples from the prior (but only when it is Gaussian). The primary contribution is the new CW statistical distance, which is the l2 distance between projected distributions, integrated over all possible projections (although not calculated as so in practice).

Plugging this distance into the WAE produces similar performance to existing WAE variants, but does not really advance the existing achievable performance.  Overall, I quite liked the paper and think it is well-written, but I believe the authors need to highlight at least one practical advance introduced by the CW distance (in which case I will raise my score). Some potential options include:

1) Faster training times. It seems to me one potential advantage of the closed-form distance would be that the stochastic WAE-optimization can converge faster (due to lower-variance gradients).  However, the authors only presented per-batch processing times as opposed to overall training time for these models.

2) Stabler training. Perhaps sampling from the prior (as needed to compute statistical distances in the other WAE variants) introduces undesirable extra variance in the training procedure. The authors could run each WAE training process K times (with random initialization) to see if the closed-form distance enables more stable results.

3) Usefulness of the CW distance outside of the autoencoder context.
Since the novelty of this work lies in the introduction of the CW distance, I would like to see an independent evaluation of this distance as a  general statistical distance measure (independently of its use in CWAE). Can you use this distance as a multivariate-Gaussian goodness of fit measure for high-dimensional data drawn from both Gaussian and non-Gaussian distributions and show that it actually outperforms other standard statistical distances (e.g. in two-sample testing power)?

Without demonstrating any practical advance, this work becomes simply another one of the multitude of V/W-AE-variants that already exist.

Other Comments:

- While I agree that standard WAE-MMD and SWAE require some form of sampling to compute their respective statistical distance, a variant of WAE-MMD could be converted to a closed form statistical distance in the case of a Gaussian prior, by way of Stein's method or other existing goodness-of-fit measures designed specifically for Gaussians. See for example:

Chwialkowski et al: https://arxiv.org/pdf/1602.02964.pdf

which like CW-distance is also a quadratic-time closed-form distance between samples and a target density.

Besides having closed form in the case of a Gaussian prior (which other statistical distances could potentially also achieve), it would be nice to see some discussion of why the authors believe their CW-distance is conceptually superior to such alternatives.

- Silverman's rule of thumb is only asymptotically optimal when the underlying data-generating distribution itself is Gaussian. Perhaps you can argue here that due to CLT: the projected data (for high-dimensional latent spaces) should look approximately Gaussian?

After reading the revision: I have raised my score by 1 point and recommend acceptance.

---

> ### Author Response · Authors · 2018-11-23
> **We thank the reviewer for a thorough inspection of our paper and a set of very valuable questions and suggestion.**
>
> The reviewer observed that “authors need to highlight at least one practical advance introduced by the CW distance” and suggested the following potential options:
>
> 1) Faster training times.
> 2) Stabler training.
> 3) Usefulness of the CW distance outside of the autoencoder context.
>
> Ad. 1) (faster training) The experiments show, that CWAE model approaches best generalization, measured with the FID score, much more rapidly, than it is the case with WAE or SWAE models. E.g., when trained on the CelebA problem, the FID-score in case of CWAE drops below 100 after only about 75 batches, while for the WAE model only near 400 batches, it does so. The same applies to SWAE. Needless to say, the FID score for CWAE is near a common best value (after about 500 epochs) of about 95 (these are results are for a DeConv encoder-decoder architecture, see. Appendix E for details; for a direct comparison with Tolstikhin at al’s paper results for an identical architecture to theirs are given in Table 1 in the paper) after a much shorter processing time. This is both thanks to the quicker convergence, but also due to faster batch processing (as it was shown in the paper). The MMD-like cloud-to-cloud formula for CW-distance (see equation (3) in the paper) is much more cumbersome than the actual one cloud-to-distribution used in the experiments derived in the paper and shown in equation at page 5 of the paper. The proposed Cramer-Wold kernel behaves correctly. We have added graphs describing this to the paper, exchanging those on page 8 (as the new are much more clearer). Graphs comparing CWAE, WAE and SWAE learning, on both CelebA and CIFAR10 datasets, shall be added to the Appendix.
>
> Ad. 2) (stable training) We have run repeated experiments with different initializations for all the generative models, as the reviewer has suggested. All experiments show that CWAE learning process is stable and repetitive: the standard deviations, for most of the coefficients computed during training are smaller than those of WAE or SWAE models (in particular CWAE minimizes WAE distance faster then WAE-MMD). We have added appropriate graphs to the paper.
>
> Ad. 3) (CW usefulness) We have verified how the Cramer-Wold metric works as a Gaussian goodness of fit, however, the results were not satisfactory. The tests based on Cramer-Wold metric were, in general, in the middle of compared tests (Mardia, Henze-Zirkler and Royston tests). We doubt it can be efficiently applied in this direction. However, since Cramer-Wold metric is defined by characteristic kernel, it can be applied in the large field of kernel-based methods in machine learning (where its particular advantage lies in the fact that it can be efficiently computed for the mixture of radial Gaussians).
>
> The reviewer noted that “besides having closed form in the case of a Gaussian prior (which other statistical distances could potentially also achieve), it would be nice to see some discussion of why the authors believe their CW-distance is conceptually superior to such alternatives.
>
> In our opinion sliced approach works well for neural networks, as the neural networks see/process data by applying similar one dimensional projections. Also the success of neural networks based on the classical activation functions, as compared to RBF networks, supports this.
> Concerning the closed-form, Cramer-Wold kernel is the only known to the authors, which is given by the sliced approach and has a closed form for radial gaussians.
>
> The reviewer also noted, that “Silverman's rule of thumb is only asymptotically optimal when the underlying data-generating distribution itself is Gaussian. Perhaps you can argue here that due to CLT: the projected data (for high-dimensional latent spaces) should look approximately Gaussian?”.
>
> In our opinion the model works well due to the fact that we compare it to the Gaussian N(0,I), where the Silverman’s kernel is optimal. However, if the prior in general would not be standard Gaussian, the situation could possibly be different.

---

> > ### Comment · AnonReviewer2 · 2018-11-25
> > **Willing to raise my score**
> >
> > I believe the authors have now sufficiently demonstrated some practical advantages of their CWAE model.  I will raise my score by 1 point to recommend acceptance once the authors make the following final adjustments:
> >
> > 1) Include something along the lines of this statement somewhere in the paper/appendix:
> >
> > "We have verified how the Cramer-Wold metric works as a Gaussian goodness of fit, however, the results were not satisfactory. The tests based on Cramer-Wold metric were, in general, in the middle of compared tests (Mardia, Henze-Zirkler and Royston tests)."
> >
> > to indicate that CW metric is not ideal for Gaussian goodness of fit testing.
> >
> > 2) Edit the following statement:
> > "If R has standard deviation close to one, the asymptotically optimal choice of γ is given by the Silverman’s rule of thumb"
> > which is only correct when the underlying data-generating distribution is Gaussian.
> > This needs to be explicitly stated, otherwise the statement is wrong.

---

> > > ### Author Response · Authors · 2018-11-26
> > > **We thank the reviewer for valuable comments.**
> > >
> > > We fully agree with both comments, and have inserted them in the text: the first in one before the last paragraph in Section 5, bottom of page 8 (Experiments), and the second in paragraph “Cramer-Wold distance”, Section 2 (Cramer-Wold distance:construction), page 3.

---

> > > > ### Comment · AnonReviewer2 · 2018-11-28
> > > > **Score raised**
> > > >
> > > > After reading the revision: I have raised my score by 1 point and recommend acceptance.

---

### Official Review · AnonReviewer1 · 2018-11-06
**A variation on the Wasserstein Auto-Encoders proposing a specific choice of the divergence penalty.**

**Rating:** 7
**Confidence:** 4

**Review:**

The paper introduces a novel regularized auto-encoder architecture called the Cramer-Wold AutoEncoders (CWAE). It's objective (Eq. 7) consists of two terms: (i) a standard reconstruction term making sure the the encoder-decoder pair aligns nicely to accurately reconstruct all the training images and (ii) the regularizer, which roughly speaking requires the encoded training distribution to look similar to the standard normal (which is a prior used in the generative model being trained). The main novelty of the paper is in the form of this regularizer. The authors introduce what they call "the Cramer-Wold distance" (for definitions see Theorems 3.1 and 3.2) which is defined between two finite sets of D-dimensional points. The authors provide empirical studies showing that the proposed CWAE method achieves the same quality of samples (measured with FID scores) as the WAE-MMD model [1] previously reported in the literature, while running faster (by up to factor of 2 reduction in the training time, as the authors report).

While on the AE model / architecture side I feel the contribution is very marginal, I still think that the improvement in the training speed is something useful. Otherwise it is a nicely written and polished piece of work.

Detailed comments:
(1) My main problem with this paper is that the novel objective proposed by the authors in Eq. 7 is equivalent to the objective of WAEs appearing in Eq. 4 of [1] (up to a heuristic of applying logarithm to the divergence measure, which is not justified but meant to "improve the balance between two terms", see footnote 2), where the authors use the newly introduced Cramer-Wold divergence as a choice of the penalty term in Eq. 4 of [1].
(2) When viewed in this way, CW-distance introduced in Eq (2) closely resembles the unbiased U-statistic estimate of the MMD used in WAE-MMD [1, Algorithm 2]. In other words: it may be the case that there is a choice of a reproducing kernel k such that Eq. 2 of this paper is an estimate of MMD_k between two distributions based on the i.i.d. samples X and Y. Note that if it is indeed the case, this corresponds to the V-statistic and thus biased: in U-statistic the diagonal terms (that is i = i' and j = j' in forst two terms of eq 2) would be omitted. If this all is indeed the case, it is not surprising that the numbers the authors get in the experiments are so similar to WAE-MMD, because CWAE would be exactly WAE-MMD with a specific choice of the kernel.
(3) The authors make a big deal out of their proposed divergence measure not requiring samples from the prior as opposed to WAE-MMD. However, WAE-MMD does not necessarily need to sample from the prior when used with Gaussian prior and Gaussian RBF kernel, because in this case the prior-related parts of the MMD can be computed analytically.  In other words, if the computational advantage of CWAE compared to WAE-MMD comes from CWAE not sampling Pz, the computational overhead of WAE-MMD can be eliminated at least in the above-mentioned setting.
(4) based on the name "CW distance" I would expect the authors to actually prove that it is indeed a distance (i.e. all the main axioms).
(5) The authors override the CW distance: first in Theorem 3.1 they define it as a distance between two finite point clouds, and later in Theorem 3.2 they redefine it as a distance between a point cloud and the Gaussian distribution.
(6) What is image(X) in Remark 4.1?

[1] Tolstikhin et al., Wasserstein Auto-Encoders, 2017.

---

> ### Author Response · Authors · 2018-11-23
> **We thank the reviewer for a deep insight into our paper.  We shall now, to the best of our abilities, answer all the doubts.**
>
> POINTS 1 AND 2 OF THE REVIEW
> The reviewer has noticed that the cw-distance resembles that of a U-statistic MMD estimate, and thus the proposed model very much resembles MMD itself.
>
> We fully agree with the reviewer, that CWAE is a model based on the kernel as the divergence measure for distributions, and consequently can be seen as a modified variant of WAE-MMD (we have added the respective comments in the paper, see the extended introduction, and added a section B in the appendix, which discusses the comparison in more details). However, there are some important, in our opinion, differences between those two models, which also result in an improved training speed and stability of CWAE compared to WAE-MMD (see refined experiments in section 5, as well as figures in the appendix showing comparisons between proposed CWAE and WAE and SWAE models in the Appendix). The differences are:
> Due to the properties of the constructed Cramer-Wold kernel, we are able to substitute in the distance the sample estimation d(X,Y) of d(X,N(0,I)) given by its exact formula. Consequently, the CWAE has, while being trained, potentially less stochastic perturbation then WAE-MMD.
> CWAE, as compared to WAE-MMD, has no parameters (while WAE-MMD has two). We observed that in many cases (like log-likelihood), the logarithm of the probability function works better, since it increases the role of examples with low-probability. Thus, instead of using an additional weighting parameter lambda (as in WAE-MMD) whose aim is to balance the MSE and divergence terms, we decided to automatically (independently of dimension) balance the two terms of the loss function, by taking the logarithm of the divergence. Moreover, since our kernel is naturally introduced with the sliced approach and kernel smoothing, the choice of regularization parameter is given by the Silverman's rule of thumb, and depends on the sample size (contrary to WAE-MMD, where the parameters are chosen by hand, and in general do not depend on the sample size). The appropriate clarifications are given in the appendix B.
> Summarizing, in the proposed CWAE model, contrary to WAE-MMD, we do not have to choose parameters. Additionally, since we do not have the noise in the learning process given by the random choice of the sample from normal density,  CWAE in generally learns faster than WAE-MMD, and has smaller dispersion of the cost-function during the learning process (see Figures 7 and 8, Appendix F).
>
> POINT 3
> The reviewer notices that the WAE-MMD does not need to sample when used with Gaussian prior and a Gaussian RBF kernel.
>
> We fully agree that the gaussian kernel has the close formula for the product of two gaussians. However, the problem (see Tolstikhin et al’s paper Wasserstein auto-encoders, https://arxiv.org/pdf/1711.01558.pdf, Section 4, also Bińkowski et al,  https://arxiv.org/pdf/1801.01401.pdf) that Gaussian kernel does not work well with the model, as its derivatives decrease too fast, and the model with Gaussian kernel is unable to learn to modify points which lie far from the center. We have added a respective comment in Appendix A. As to the best knowledge of the authors, the introduced Cramer-Wold kernel is the unique characteristic kernel which has the closed form for spherical gaussians, and does not have exponential decrease of derivative (as the case of RBF kernel).
>
> POINTS 4 AND 5
> As the reviewer accurately and carefully noticed, we have not formally proved that cw-distance is a true distance, and that the definition is introduced partially: first for two clouds of points, then a distribution and a cloud.
>
> This is true. We have added a respective proof in Appendix, Section A, where also the precise mathematical construction of the general form of Cramer-Wold metric is presented. We have also added the comment at the beginning of Section 3 of the paper. We hope clarifies our unintentionally imprecise original approach.
>
> POINT  6
> The reviewer asked “what is image(X) in Remark 4.1?”
>
> By image(X) we understand the set of all possible values the random vector X can attain (we have included the footnote in Remark 4.1 explaining the notation).

---

### Official Review · AnonReviewer3 · 2018-11-14
**An interesting derivation of a new distance, but should be compared with MMD**

**Rating:** 5
**Confidence:** 4

**Review:**

This paper proposes the Cramer-Wold autoencoder. The first contribution of the paper is to propose the Cramer-Wold distance between two distributions based on the Cramer-Wold Theorem. More specifically, in order to compute the Cramer-Wold distance, we first find the one dimensional projections of the distributions over random slices, and then compute the average L2 distances of the kernel density estimates of these projections over random slices. The second contribution of the paper is to develop a generative autoencoder which uses the Cramer-Wold distance to match the latent distribution of the data to the prior distribution.

While I found the derivation of the Cramer-Wold distance interesting, the final form of this distance (Eq. 2), to me, looks very similar to the MMD with a particular kernel. My main question is that: what is the main advantage of the Cramer-Wold distance to an MMD with a proper kernel?

The paper points out that the main theoretical contribution is that in the case of the Gaussian distribution, the Cramer-Wold distance has a closed form. However, I believe this is also the case in the MMD, since if one of the distributions is Gaussian or analytically known, then E[k(x,x')] in the MMD can be analytically computed.

The paper further uses this closed form property of the Cramer-Wold distance to propose the Cramer-Wold autoencoder with Gaussian priors. My question here is that how is this method better than the standard VAE, where we also have an analytic form for the ELBO when the prior is Gaussian, an no sampling is required. Indeed, in VAEs, the prior does not have to be Gaussian, and as long as the density of the prior can be evaluated, we can efficiently optimize the ELBO without sampling the prior; which I don't think is the case for the Cramer-Wold autoencoder. I believe the main advantages of methods such as WAE is that they can impose priors that do not have exact analytic forms.

---

> ### Author Response · Authors · 2018-11-23
> **We thank the reviewer for a thorough inspection of our paper and a set of very valuable questions and suggestion.**
>
> We thank the reviewer for insight into our paper. The reviewer found some points, where we were not clear enough. It is now the time to respond to them.
>
> 1. The reviewer noticed, that  “in the MMD, since if one of the distributions is Gaussian or analytically known, then E[k(x,x')] in the MMD can be analytically computed”.
>
> According to the best knowledge of the authors, the Cramer-Wold kernel (which defines the Cramer-Wold metric), except for the classical RBF kernel, is the only known characteristic kernel which has closed form for radial gaussians, and we believe the respective computations in other cases (like the inverse quadratic kernel used in WAE-MMD), would be highly nontrivial.
>
> 2. The reviewer also points out, that the evidence lower bound ELBO, when used with a notiGaussian prior results in case of VAE in a generally analytic formula. It was never the intention of the authors to sneak in that VAE cannot do it. Our primary goal was to define a method for training the Gaussian prior generative model using a different closed form formula for the distribution distance. At the same time VAE requires encoder to be Gaussian non-deterministic, and random decoder, which is not the case in CWAE (as well as in a WAE model, see Tolstikhin https://arxiv.org/pdf/1711.01558.pdf). The kernel used in the derivation is not a Gaussian kernel but has a closed form formula for a product of two Gaussians (see last equation in the current paper), itself not being Gaussian. The Gaussian kernel itself is not well suited,  because it has an exponential rate of decay, and loses much information on the outliers (see also Bińkowski et al.,  https://arxiv.org/pdf/1801.01401.pdf, section 2.1). Our objective was to add a method alternative to the WAE method, but simpler in use (e.g. less parameters to be found).
>
> We have extended the contribution part (in the introduction) and added Sections A and B to the Appendix, to make things clearer.

---

> ### Author Response · Authors · 2018-12-11
> **Follow-up**
>
> Thank you again for your comments and suggestions. Have our responses and the changes we made to the manuscript addressed all of your concerns?

---

### Meta-Review · Area_Chair1 · 2018-12-14
**A nice closed-form kernel for WAE-MMD, but concerns about novelty.**

**Confidence:** 3
**Recommendation:** Reject

**Metareview:**

The reviewers in general like the idea of using the Cramer-Wold kernel, noting that its heavy tails and closed form solution are appealing properties that lead to increased stability and improved training. The main concern was novelty, as this paper can be seen as simply changing the kernel in WAE-MMD. One suggestion is to more heavily highlight the CW-distance, and in particular to find another useful application for it outside of WAE-MMD.

The paper emphasizes frequently that the closed-form loss function is a critical feature of this approach, however I don’t see any experiments that optimize WAE-MMD under the CW-distance while sampling from the Gaussian. This is important to measure the degree to which any improvement is attributable to a closed-form solution, or to the distance measure itself.